# Convolutional-Neural-Network-Based Hexagonal Quantum Error Correction Decoder

**Aoqing Li** [1], **Fan Li** [1], **Qidi Gan** [2] **and Hongyang Ma** [2,*]

1   School of Information and Control Engineering, Qingdao University of Technology, Qingdao 266033, China; aoqing0405@163.com (A.L.); lifan202202@163.com (F.L.)
2   School of Science, Qingdao University of Technology, Qingdao 266033, China; 18606377972@163.com
*   Correspondence: hongyang_ma@aliyun.com

**Abstract:** Topological quantum error-correcting codes are an important tool for realizing fault-tolerant quantum computers. Heavy hexagonal coding is a new class of quantum error-correcting coding that assigns physical and auxiliary qubits to the vertices and edges of a low-degree graph. The layout of heavy hexagonal codes is particularly suitable for superconducting qubit architectures to reduce frequency conflicts and crosstalk. Although various topological code decoders have been proposed, constructing the optimal decoder remains challenging. Machine learning is an effective decoding scheme for topological codes, and in this paper, we propose a machine learning heavy hexagonal decoder based on a convolutional neural network (CNN) to obtain the decoding threshold. We test our method on heavy hexagonal codes with code distance of three, five, and seven, and increase it to five, seven, and nine by optimizing the RestNet network architecture. Our results show that the decoder thresholding accuracies are about 0.57% and 0.65%, respectively, which are about 25% higher than the conventional decoding scheme under the depolarizing noise model. The proposed decoding architecture is also applicable to other topological code families.

**Keywords:** quantum error correction; heavy hexagonal code; convolutional neural network decoder





## 1. Introduction

Like classical computers, qubits are an important part of quantum computers [1–3]. The difference between qubits and classical bits is that they have quantum mechanical properties, such as entanglement and superposition, which enable quantum computers to solve complex problems that cannot be solved by classical computers. Refs. [4–6]. However, quantum states are very fragile and susceptible to external environmental noise, leading to inevitable interactions between qubits and their environment, which can cause a decoherence and errors in qubits [7–9]. The development of efficient quantum error-correcting codes is the key to overcoming this problem and is an important guarantee for quantum information transmission. Quantum error correction operates similarly to classical bit error correction and centers on the use of redundant qubits to encode the information of one qubit into multiple entangled qubits [10,11]. These entangled qubits are called logic qubits and have superior properties compared to single qubits.

The principle of quantum error-correcting codes is to reduce the effect [12] of noise on qubits by encoding physical qubits into logical qubits. Various quantum error-correcting codes have been proposed and studied, of which surface codes are the main area of research today. Surface codes are a family of quantum error-correcting codes that utilize neighborhood interactions between planarly arranged physical qubits for error correction purposes [13–15]. However, on the dense lattice of a surface code, each qubit is connected to four other qubits, potentially causing multiple frequency collisions between them. To solve this problem, researchers have proposed a new error-correcting code, the heavy hexagonal code [16]. This code is based on a combination of degree-two and degree-three vertices connecting the vertices of physical qubits. Thus, it can be viewed as a fusion of

surface codes and Bacon–Shor codes [17]. One of the main advantages of this approach is that it reduces the number of different frequencies required for the implementation, while minimizing qubit crosstalk by introducing auxiliary qubits or flagged qubits in the syndromic measurement. This coding structure fits well with the hexagonal structure of a quantum computer.

Another major research focus in quantum error correction is how to achieve a fast and stable decoding of error-correcting codes, and the threshold is a key metric for evaluating decoder performance. The heavy hexagonal code [18] utilizes tight interactions between physical qubits to achieve error correction, but it poses considerable challenges for decoding. Currently, the research schemes for decoding heavy hexagonal codes mainly use the minimum-weight perfect-matching algorithm (MWPM) [19], which has an X error threshold of about 0.45%. In addition to this, the perfect-matching decoder and the maximum likelihood decoder have been used to reduce the logic error rate, respectively, in Ref. [20]. In this paper, we propose to decode heavy hexagonal codes using a machine-learning-based CNN architecture and also consider the application of machine learning in decoding topological quantum error-correcting codes [21,22]. Under a depolarizing noise model, we attempt to reach the error correction threshold for heavy hexagonal coding. Our numerical results show that the decoding threshold can be reached around 0.57% under the CNN architecture. As the decoding distance increases, more complex neural network models are required. Based on this, we optimized the CNN architecture to further investigate the decoding performance of heavy hexagonal coding. The decoding threshold of the optimized heavy hexagonal coding is about 0.65%, which is close to the threshold of 0.67% for standard surface coding. Compared to the MWPM decoding algorithm, our decoding method improves the decoding threshold by about 25%.

The main points of this paper's work are as follows. In Section 2, we introduce the basic concepts of deformed heavy hexagonal codes in topological surface codes. In Section 3, we analyze the error measure of the heavy hexagonal code and propose a machine learning convolutional-neural-network-based decoding scheme for the code. Section 4 discusses the performance and thresholding analysis of the decoder. Finally, Section 5 gives the conclusions.

## 2. Background

A surface code is a type of stabilizer code whose layout is created by arranging and combining four-dimensional square matrices defined on a two-dimensional plane [23]. In the structure of a heavy hexagonal code, the degrees of the qubits are usually two (i.e., they can only interact with two qubits) and three, with most of them being of degree two. Heavy hexagonal codes are a hybrid of surface and Bacon–Shor codes [24] and significantly increase the average qubit density while retaining qubits of degree four compared to the surface code structure. The lattice of heavy hexagonal codes is used to encode logical qubits, with "heavy" indicating that the qubits are located at the vertices and edges of the hexagonal lattice.

The heavy hexagonal code is a special type of subsystem code [25] that combines elements from both the surface code and the Bacon–Shor code. Subsystem codes can be defined through the use of stabilizer groups, which are mathematical constructs that describe how to detect errors and preserve quantum information. The stabilizer group can be represented by the tensor product of n one-dimensional Pauli operators.

$$P_n = \{w_1 \otimes w_2 \otimes \cdots \otimes w_n : w_i \in \{I, X, Y, Z\} (1 \leq i \leq n)\}. \tag{1}$$

Like other error-correcting codes, heavy hexagonal codes are also defined by a set of gauge operators, denoted by $\mathcal{G}$. Since the code space in a subsystem code consists of multiple equivalent subsystems [26], the role of the gauge operator is to bring a codeword to an equivalent subsystem. In the heavy hexagonal code structure, logical information is encoded and protected in a subsystem, and the Hilbert space of the entire codeword will be expanded into a larger Hilbert space, which is beneficial to the protection subsystems that

are immune to noise [27]. The gauge operators for the heavy hexagonal code are similar to other quantum error-correcting codes and are also represented by Pauli operators. Next, we give the gauge operator definitions for the heavy hexagonal code:

$$
\mathcal{G} = \langle Z_{i,j}Z_{i+1,j}, X_{i,j}X_{i,j+1}X_{i+1,j}X_{i+1,j+1},
$$
$$
X_{1,2m-1}X_{1,2m}, X_{d,2m}X_{d,2m+1} \rangle. \tag{2}
$$

Here, $i$ and $j$ are used to represent the row index and column index, respectively, and $i,j = 1,2,\ldots,d\text{-}1, m = 1,2,\ldots,(d-1/2)$. The constraint is that $i + j$ is even for the second term. Figure 1 is a schematic diagram of the lattice layout of the heavy hexagonal code with $d = 5$ and the controlled-NOT (CNOT) gates coding for the syndrome measurement. As shown in the figure, it is mainly composed of two areas, where the red areas are composed of weight-four $X$-type gauge generators and weight-two $X$-type gauge generators (red area), and the blue areas are composed of weight-two $Z$-type gauge generators (blue area). The $X$-type and $Z$-type gauge generators are used to correct for bit flips and phase errors, respectively.

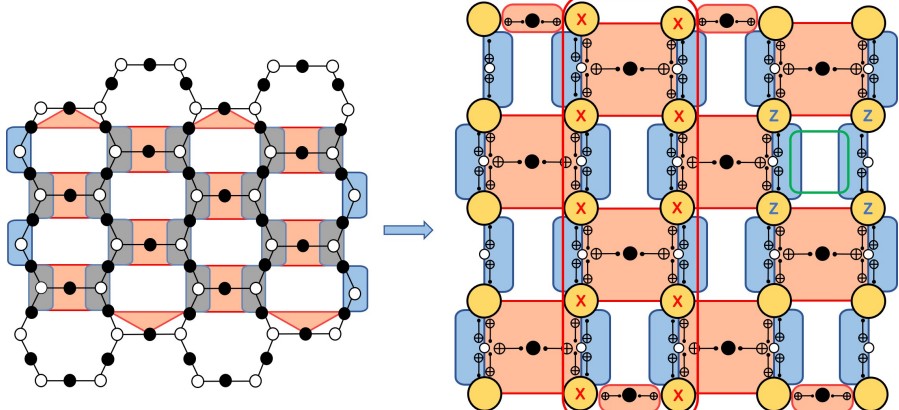

**Figure 1.** The left side of the figure shows a schematic diagram of the lattice layout of the heavy hexagonal code, while the right side displays the square lattice structure layout of the heavy hexagonal code with $d = 5$. The syndrome error measurement is carried out between each qubit through CNOT gates scheduling. Yellow vertices represent data qubits, white vertices are flag qubits, and dark vertices represent ancillary qubits used to measure the $X$-type gauge generators (red areas) and $Z$-type gauge generators (blue areas).

As previously stated, the heavy hexagonal code belongs to the family of stabilizer codes, and its stabilizer group is defined as follows:

$$
\mathcal{S} = \langle Z_{i,j}Z_{i,j+1}Z_{i+1,j}Z_{i+1,j+1}, Z_{2m-1,d}Z_{2m,d},
$$
$$
Z_{2m,1}Z_{2m+1,1}, \Pi_i X_{i,j}X_{i,j+1} \rangle, \tag{3}
$$

where $Z_{i,j}Z_{i,j+1}Z_{i+1,j}Z_{i+1,j+1}$ is a weight-four stabilizer. The measurement outcome of the stabilizer is the product of the measured eigenvalues of the two weight-two gauge generators $Z_{i,j}Z_{i+1,j}$ and $Z_{i,j+1}Z_{i+1,j+1}$, with the constraint that $i + j$ is even for the first term. As shown in Figure 1, the left and right boundaries of the heavy hexagonal code are composed of weight-two surface-type stabilizer operators, and the top and down boundaries are composed of Bacon–Shor type stabilizers. The middle areas are mainly composed of bulk operators with a weight of four. The $X$-stabilizer operator consists of two vertical columns of $X$-type strips, which can be measured by the product of the measured eigenvalues of all weight-four bulk $X$-type gauge generators and weight-two boundary $X$-type gauge generators. The $Z$-stabilizer operator consists of four $Z$-type gauge generators in the bulk and two $Z$-type gauge generators on the left and right boundary. All operators in the gauge operator $\mathcal{G}$ are commutative with the set of stabilizer operators $\mathcal{S}$. However,

different Pauli types on overlapping gauge operators do not necessarily commute. Thus, qubit errors can be inferred by judging the stabilizer eigenvalues [28–30].

## 3. Designing a Machine-Learning-Based Decoder for the Heavy Hexagonal Code

### 3.1. Error Correction

Surface codes and Bacon–Shor codes are the most promising error-correcting codes for building quantum computers. Stabilizer formalism is an important basis for understanding the heavy hexagonal code. The hexagonal lattice code is a quantum error-correcting code that maps $k$ logical qubits to n physical qubits and uses a $2^k$-dimensional code space $\mathcal{H}_C$ to accommodate these encoded qubits. The code space can be viewed as a $2^k$-dimensional subspace of the $2^n$-dimensional Hilbert space defined by codewords $|\psi\rangle$,

$$\mathcal{H}_C = \{|\psi\rangle \in \mathcal{H} \mid g|\psi\rangle = |\psi\rangle \forall g \in \mathcal{S}\}. \tag{4}$$

The stabilizer group $\mathcal{S}$ is an Abelian subgroup consisting of the Pauli group $P_n$ of $n$-qubits. $\mathbb{Z}(\mathcal{S})$ is the set of Pauli group elements that satisfy $\mathbb{Z}(\mathcal{S})= \{P \in P_n \mid PS = SP \mid \forall S \in \mathcal{S}\}$. The stabilizer code $\mathbb{C}(\mathcal{S})$ is composed of the minimum number of $m$ generators $S_j$.

$$\mathcal{S} = \langle S_1, S_2, \ldots S_m \rangle, \tag{5}$$

and each encoded qubit has a pair of logical operators $(\overline{X}_j, \overline{Z}_j)$,

$$\overline{X}_j, \overline{Z}_j \in \mathbb{Z}(\mathcal{S}) \setminus \mathcal{S}, \qquad \overline{X}_j \overline{Z}_j = -\overline{Z}_j \overline{X}_j. \tag{6}$$

Then, we perform a stabilizer measurement for each generator $S_j$ in $\mathcal{S}$ and define the syndrome vector $s \in Z_2^m$ as

$$s = (s_1, s_1 \ldots s_m) \; where \; s_j = \begin{cases} 0, & if \; measurement \; of \; S_j \; yields \; +1. \\ 1, & otherwise \end{cases} \tag{7}$$

The heavy hexagonal code can be thought of as a hybrid of surface codes and Bacon–Shor codes. Therefore, we can use the error correction properties of surface codes and Bacon–Shor codes to perform error correction on the heavy hexagonal code. In the hexagonal code error, the *X* and *Z* errors can be corrected by the decoding process of the surface code and the Bacon–Shor code, respectively, where the *X*-error correction corresponds to a classical topological surface code [31,32]. Figure 2 is a circuit diagram for measuring the gauge operators of *X*-type and *Z*-type. CNOT gates are one of the most commonly used gates in quantum computing for implementing interactions between quantum bits. In CNOT scheduling, we try to optimize the order of operation of CNOT gates to minimize the total number of error positions in the synthesis measurement. A single error correction cycle requires 11 time steps (7 for *X* and 4 for *Z*), which include qubit initialization and measurement. As shown, the heavy hexagonal code have deeper measurement circuits compared to topological surface codes, but utilizing flag qubits can correct weight-two errors caused by a single error during measurements of the weight-four *X*-gauge. This is also a major advantage of the heavy hexagonal code.

In this paper, in order to simplify the noise model, we combine the depolarization noise model with the heavy hexagonal code to verify the error correction performance of the heavy hexagonal code. For simpler noise models, the processing of the depolarized noise model is independent. Given a quantum state $\rho$, the evolution of the state in the depolarization noise model is:

$$\rho \rightarrow (1 - p_x - p_y - p_z)\rho + p_x X \rho X^\dagger + p_y Y \rho Y^\dagger + p_z Z \rho Z^\dagger, \tag{8}$$

and $p_x = p_y = p_z$. Among them, $p_x$, $p_y$, and $p_z$, respectively, represent the probability of different Pauli *X*, *Y*, and *Z* errors. In the heavy hexagonal code measurement circuit, we say that the flag qubit is flagged when it has a nontrivial measurement result. Note

that, nontrivial flag results indicate a weight-two or weight-four data qubit error occurred, or a measurement error triggered the flag. The measurement syndrome is where the flag qubit turns out to be nontrivial in the *X*-type gauge operator measurements when *X*-errors occur on qubits, and the measurement of erroneous data qubits occurs during *Z*-stabilizer measurements. Similar to *X*-error, the *Z*-error captures the measurement syndrome of the data qubit by judging the flag qubit's measurement results, thereby inferring the data qubit where the error occurred, and finally, by looping the measurement circuit to extract the syndrome information and analyze the syndrome information to correct errors that occurred on the qubit. However, due to the complexity of the combination of syndrome information, it is difficult for us to select them with commonly used statistical methods. Therefore, we now propose a machine-learning-based CNN decoder [33,34] where we exploit the translation invariance of the heavy hexagonal code to feed error syndromes to a convolutional neural network for a faster determination of optimal syndromes and achieve the maximum good threshold by optimizing the condition [35]. In this paper, we focus on the degree of local correlation of error syndromes depending on the distance of the heavy hexagonal code [36].

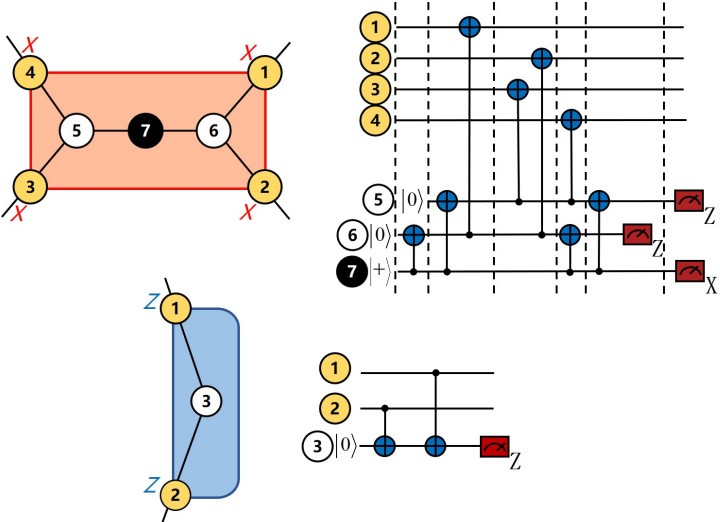

**Figure 2.** The figure displays the circuit diagram for *X*-type and *Z*-type parity check measurements of the heavy hexagonal code. Yellow circles represent data qubits, black circles represent measurement qubits, and white circles represent flag qubits. Two flag qubits (white circles) are used to measure weight-four *X*-type gauge generators, while one flag qubit is used to measure the weight-two *Z*-type gauge generators.

### 3.2. Convolutional-Neural-Network-Architecture-Based Decoder

A convolutional neural network is a type of artificial neural network, usually used for data classification problems. In machine learning [37,38], we can reduce the decoding problem to a classification problem. It is a deep learning algorithm mainly composed of three structures: convolution, pooling, and activation. The characteristics of the neural network are defined by the convolutional layer, and the error information between qubits can be transmitted and maintained through the convolutional layer. The two-dimensional convolution operation is defined as follows:

$$z_{ij} = \sum_{k=l=1}^{f,f} w_{kl} x_{k'l'}, \tag{9}$$

$$k' = k + i - \frac{f+1}{2}, l' = l + j - \frac{f+1}{2}, \tag{10}$$

$z_i$ represents the feature map. $w$ is an array of weights that make up the filter, and $x$ represents the input matrix. $k$ denotes the row index of the input matrix, which is used to traverse the elements of the input matrix. Due to the grid layout of the heavy hexagonal code and the correlation between qubit vertices, CNN is considered as a decoding method for the heavy hexagonal code [39,40]. Additionally, the syndrome measurement information of the heavy hexagonal code will serve as the input of the fully connected neural network. The ideal error recovery operator can be obtained by mapping the input feature space through the fully connected layer to achieve classification accuracy. We utilize the topological properties of the heavy hexagonal error correction code to map the heavy hexagonal-coded image data into the CNN input, and for each hexagonal vertex lattice, it is mapped to the corresponding position in the 2D square lattice according to its coordinate information. In the convolutional layer, a certain size of convolutional kernel is selected to perform the convolutional operation on the input data, and the output value is used to determine whether the classification is correct or not; if the output value is +1, it means that the output layer is classified correctly, and if the output value is −1, it means that the classification is wrong.

In our experiment, we first train the neural network with smaller data and coding distance, and then expand the coding distance when the training threshold is close to our desired optimal threshold [41,42]. However, as the coding distance increases, the training time becomes longer and the performance requirements of the machine become higher. Therefore, various conditions of the CNN network structure must be optimized to reduce the cost of training. The weights in the neural network play a decisive role in the training results. As shown in Figure 3, the convolutional neural network can use the backpropagation algorithm to adjust the network weights to achieve the training results.

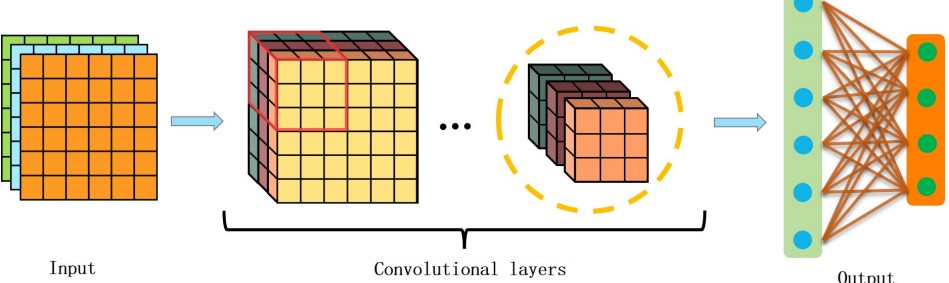

Input                    Convolutional layers                  Output

**Figure 3.** The figure shows the various components and processes involved in the algorithm and how they interact with each other. The topological properties of the code are utilized to map the heavy hexagonal-coded image data into the CNN input and a new feature map is generated by performing convolution operations using a set convolution kernel. Finally, the trained classification or prediction results are available at the output layer.

As we know, machine learning methods are mainly divided into supervised learning and unsupervised learning [43–45]. In our CNN model, the convolutional neural network is mainly used to supervise randomly generated errors and their corresponding $X$ and $Z$ stabilizer information and infer the qubit and error type of the error based on the stabilizer information. Considering that the categorical cross-entropy loss function performs well in classification problems, it is consistent with our neural network. Therefore, we used the classification cross-entropy loss function to ensure that each classification cross-entropy value was as close to the true value as possible. To avoid local optimizers and improve parameter adjustment speed and accuracy during gradient descent after each iteration, we used the currently popular Adam optimizer. When the quantity of data and the number of iterations increased, in order to solve the problem of gradient dispersion in the backpropagation process, we adopted the RestNet [46] structure in the CNN network. The main role of ResNet networks is to solve the problem of gradient vanishing in network training by connecting across layers. Specifically, ResNet networks build deep networks

by introducing residual blocks. Each residual block consists of two or three convolutional layers that include a cross-layer join. The cross-layer connection adds the input feature map directly to the output of the residual block, thus forming the residuals. In this way, the network learns the changes in the residuals instead of learning the mapping relationship between the inputs and outputs directly. The output of the RestNet network can be represented as:

$$y = F(x, \{W_i\}) + x. \tag{11}$$

where $F(x, \{W_i\})$ represents the residual function, and $\{W_i\}$ represents the weight parameters learned inside the residual function. This defines the error between the predicted value and true value of the neural network at this moment.

$$L = -\sum_x q(x) \log r(x), \tag{12}$$

$q(x)$ represents the true predicted result, and $r(x)$ represents the output result. The ResNet architecture can build deeper models by using residual functions to obtain the residual blocks at 7 and 14 iterations. Moreover, it reduces the dimension of our convolution operation, increases the iteration speed, and further reduces the convolution kernel dimension to ensure a reduced network depth and achieve better convolutional effects.

*3.3. Training*

The objective of training the dataset through the CNN model is to obtain a syndrome with an error rate below the threshold, which can better predict and correct the model to complete the error correction. This is a significant challenge for the neural network model when the error rate of the syndrome is high. Therefore, in model training, we trained the neural network with the configuration corresponding to an error rate close to the threshold, which can enhance the accuracy of the model.

In our work, we first chose the RestNet7 network architecture to perform small code-distance training on the collected dataset and generated a prediction model on this dataset with a low error rate.

$$C = \sum_{i=1}^{N} Y_i \times [\log(G * (X_i)) + (1 - Y_i)] \times \log(1 - G * (X_i)), \tag{13}$$

where $N$ represents the total quantity of data, $GX_i$ represents the true class label, and $Y_i$ represents the predicted probability returned by the neural network. To further verify the error correction performance of the heavy hexagonal code, we increased the code distance. Correspondingly, the dataset also increased, so we adjusted the network architecture and used RestNet14 to train the increased code distance. Figure 4 shows a schematic diagram of the iterative training of the two network architectures of RestNet7 and RestNet14. It can be seen from the figure that the training accuracy of the RestNet14 network architecture is better than that of RestNet7 and can achieve an ideal accuracy of more than 96%. Table 1 provides the training dataset, iteration steps, and accuracy of the MWPM algorithm and CNN decoder provided by us.

**Table 1.** The accuracy of the prediction model.

|  | Trainable Data | Steps | Accuracy |
|---|---|---|---|
| MWPM | $1.46 \times 10^5$ | $4.7 \times 10^4$ | 74.699% |
| RestNet7 | $1.35 \times 10^3$ | $3.3 \times 10^3$ | 84.593% |
| RestNet14 | $2.78 \times 10^3$ | $3.5 \times 10^3$ | 84.967% |

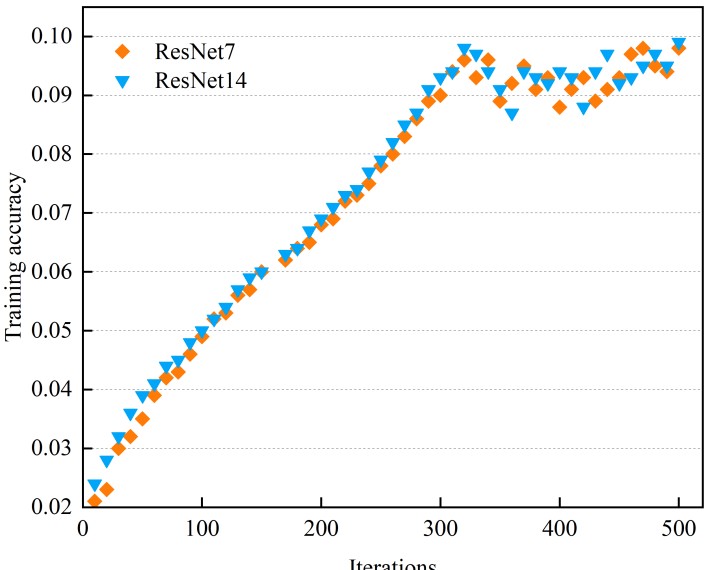

**Figure 4.** Comparison of the accuracy simulation results of RestNet7 (marked by orange diamonds) and RestNet14 (marked by blue triangles) network training iterations.

## 4. Simulation and Analysis

We used a CNN network to generate a prediction model for the heavy hexagonal code error correction. To ensure the boundary and periodicity of the grid after each convolution step, regular padding was performed in our CNN model [47]. The random error rate was represented by the physical qubit error rate and calculated from the syndrome via sampling. We provided the obtained heavy hexagonal code threshold along with a performance analysis for different code distances during training of our convolutional neural network. Furthermore, we analyzed the impact of different network layers' efficiency and accuracy on decoding threshold. Our results showed that the functional properties of different layers within the CNN model can significantly affect the decoding threshold. For example, models with more convolutional layers and filters exhibited better decoding accuracy than those with fewer layers and filters. However, increasing network depth also led to longer training times and greater computational requirements.

In the beginning, we decoded the heavy hexagonal code with distances of three, five, and seven using the MWPM algorithm and convolutional neural networks. Figure 5 shows a schematic of the logical error rate under the MWPM algorithm decoder. From the graph, it can be seen that the logical error rate increases rapidly with the increasing physical error rate and reaches the decoding threshold at 0.0040, where noise interference on quantum bits can be suppressed. After surpassing the threshold, the increase in logical error rate slows down. The threshold for the CNN-based decoding is shown in Figure 6. It can be observed from the graph that under the same code distance, the decoding threshold based on the CNN can reach 0.0057. Under the CNN model, the logical error rate generally shows a slow upward trend. Compared with the MWPM algorithm, the decoding threshold has been improved to some extent, but the threshold strength still needs to be further increased.

Increasing the code distance to nine is an effective way to improve the error correction capability of the heavy hexagonal code. In this study, we increased the code distance from three, five, and seven to nine and further compared the threshold results of the two different decoders. As shown in Figure 7, as the code distance increases, the logical error rate rises significantly, and the decoding threshold of the MWPM algorithm increases to 0.0045. Above this threshold, the rising trend of the logical error rate is almost the same for different code distances. However, the training time and dataset size grow exponentially due to the increase in code distance. Therefore, we used the RestNet14 network to predict the correction model [48–50]. As shown in Figure 8, using a CNN-based

decoding solution can reduce the logical error rate by 0.005. Specifically, when the code distance of the weighted hexagonal code is nine, the decoding threshold based on the CNN network reaches 0.0065, which is very close to the optimal threshold. This indicates that the heavy hexagonal code decoder based on machine learning technology can improve its error correction capability to some extent and performs better than the traditional MWPM decoding algorithm. Although limited by the training time and dataset size, the use of machine learning technology is still an effective method worth exploring.

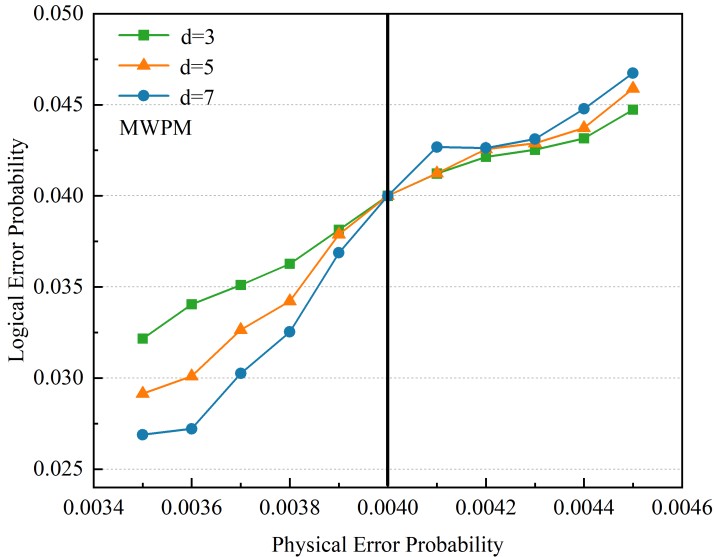

**Figure 5.** The figure shows the variation in the logic error rate of the MWPM decoder for the heavy hexagonal code at code distances of 3 (marked with green squares), 5 (marked with orange triangles), and 7 (marked with blue circles). The threshold is indicated as 0.0040.

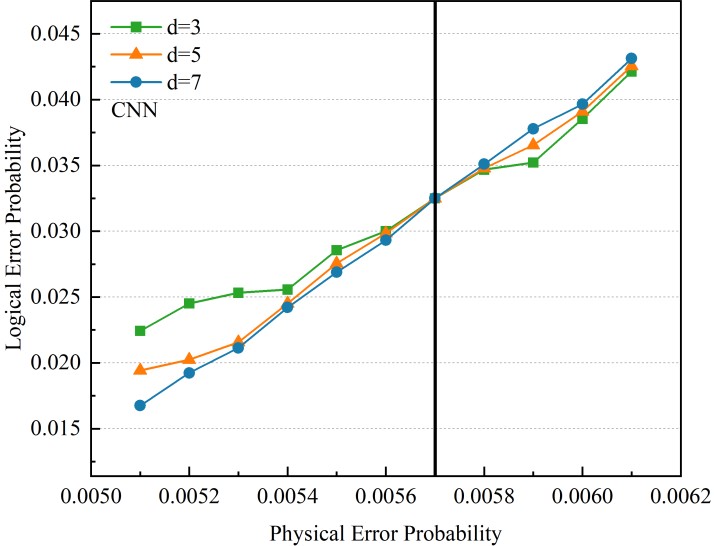

**Figure 6.** The figure displays the variation in the logic error rate of the CNN decoder for the heavy hexagonal code at different code distances. The threshold is indicated as 0.0057. As the physical error rate increases, the logical error rate also increases gradually.

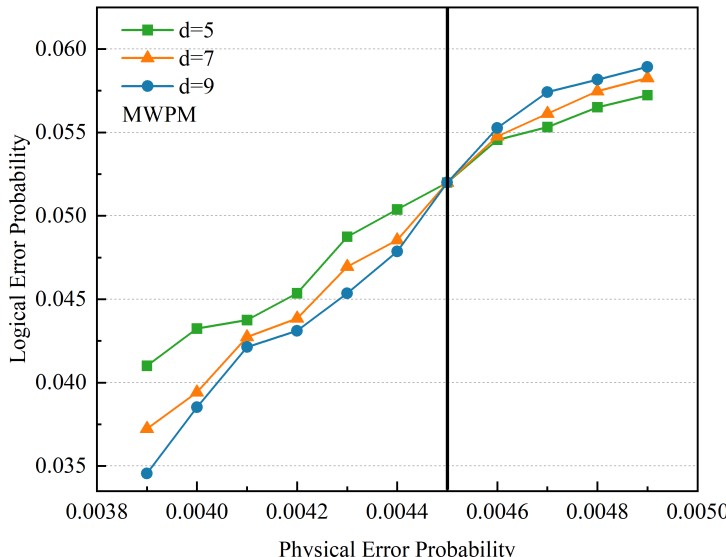

**Figure 7.** The figure shows the variation in the logic error rate of the MWPM decoder for the heavy hexagonal code at code distances of 5 (marked with green squares), 7 (marked with orange triangles), and 9 (marked with blue circles). The threshold is indicated as 0.0045.

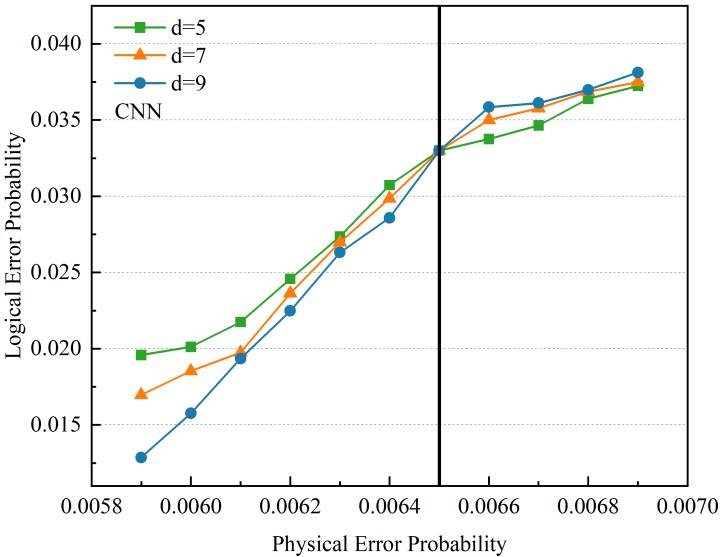

**Figure 8.** The figure displays the variation in the logic error rate of the CNN decoder for the heavy hexagonal code at different code distances. The threshold is indicated as 0.0065. The heavy hexagonal threshold is close to the optimal threshold, and the logical error rate increases gradually as the physical error rate increases.

As shown in the schematic diagram of the network iterative training above, in convolutional neural networks, the number of iterations and layers have a significant impact on model accuracy. By adjusting the number of iterations and using different network layers, we can improve training accuracy. Initially, we used the RestNet7 network for model iteration prediction. When the number of iterations exceeded approximately 300, the neural network's accuracy reached around 9.2%. Subsequently, as the number of iterations increased, the neural network's performance remained largely unchanged and was prone to overfitting. We then increased the convolutional layer to obtain more accurate training results and used the RestNet14 network to improve the convolutional performance. As shown in the figure, by comparing the training results of RestNet7 and RestNet14 networks,

we can see that prior to 320 iterations, the accuracy of the RestNet14 network was slightly better than that of the RestNet7 network, with each training interval being approximately 0.2% higher. The RestNet14 network achieved the highest training accuracy at about 320 iterations. After more than 320 iterations, the accuracy of both networks had minor fluctuations. Thus, through a comparison of the two networks' training, the RestNet14 network approached the optimal decoder's performance after full-depth training, while the decoding efficiency and accuracy were significantly improved.

## 5. Conclusions

As an essential technology in quantum computing and communication, quantum error-correcting codes have significant research value and practical applications. However, the reliability of codes is severely challenged in complex quantum environments. Meanwhile, the development and application of emerging quantum error-correcting codes also face many difficulties and challenges. In recent years, heavy hexagonal codes have been proposed as a new type of quantum error-correcting codes and gradually attracted the attention of researchers. Compared with other quantum error-correcting codes (such as surface codes and concatenated codes), heavy hexagonal codes are more accessible to implement in the laboratory and can effectively resist conventional errors.

However, the decoding process of the heavy hexagonal code still faces a series of problems. Due to the complexity of coding methods, the distances for different heavy hexagonal codes vary significantly, which poses a great challenge for the design and use of decoders. For example, the traditional MWPM can be used for low-to-moderate-distance heavy hexagonal codes but shows a considerable drop in performance at higher distances. Therefore, it is necessary to explore new algorithms and technologies to improve the error-correcting performance of the heavy hexagonal code. Therefore, this paper proposed a machine-learning-based decoder for the heavy hexagonal code, which achieved better quantum error correction performance under ideal conditions. We empirically tested the decoder using 7-layer and 14-layer residual networks for heavy hexagonal codes with distances three, five, seven, and nine. We also demonstrated that the CNN-based training of heavy hexagonal codes achieved a threshold of 0.65%, outperforming the existing MWPM decoder. However, machine-learning-based decoding algorithms still face many challenges, such as how to train and test with complex noise models, and how to use machine learning decoders in real quantum environments. In the future, we will continue to explore more efficient and accurate prediction models for quantum error correction.

**Author Contributions:** Writing—original draft, A.L.; data curation, A.L.; formal analysis, F.L. and Q.G.; methodology, A.L. and H.M. All authors have read and agreed to the published version of the manuscript.

**Funding:** This work was supported by the Natural Science Foundation of Shandong Province, China (grant no. ZR2021MF049), and Joint Fund of Natural Science Foundation of Shandong Province (grant no. ZR2022LLZ012), and Joint Fund of Natural Science Foundation of Shandong Province (grant no. ZR2021LLZ001).

**Institutional Review Board Statement:** Not applicable.

**Informed Consent Statement:** Not applicable.

**Data Availability Statement:** The data are contained within the article.

**Conflicts of Interest:** The authors declare no conflict of interest.

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
