# Peer review of "Convolutional-Neural-Network-Based Hexagonal Quantum Error Correction Decoder"

_applsci, doi:10.3390/app13179689_

Round 1

Reviewer 1 Report

The authors have chosen a convolution neural network based hexagonal quantum error correction, which is becoming increasingly popular in quantum computing. Several of the comments are listed below for your consideration.

1.      This work has not dealt with quantum computing aspects such as superposition, entanglement, and interference.

2.      Figure 2 measurement output should be discussed. Which platform you write the sample code and prove error to be recognised with proper place during decoding.

3.      The authors are unable to demonstrate novelty.

4.      The analysis is more broad than problem-specific.

5.      Only the CNOT gate is used in Figure 2, but the authors are unable to prove the logical.

6.      Discuss the hexagonal quantum error correcting decoder technique.

7.      Examine Figures 4–8 to learn about measurement tools and quantum assembly code libraries.

8.      Need to enhance quality of literature review include more references deal with quantum error control

a. "Suppressing quantum errors by scaling a surface code logical qubit." Nature 614, no. 7949 (2023): 676-681.

b. https://doi.org/10.1504/IJCAT.2017.086558

 9.      Proof that this work was relevant to both machine learning and quantum technologies.

Need to improve english mainly in abstract section. 

Author Response

Thank you for your guidance, our team for your questions in the form of a WORD document to reply line by line!

Reviewer 2 Report

The authors proposed a heavy hexagonal decoder which is optimized by a convolutional neural network outperforming minimum weight perfect matching algorithm. The proposal by the authors, although not novel, is interesting since the use of machine learning on quantum correction algorithms may reduce the current issues on superconducting architectures which are the base of many NISQ devices. However, there are some important points to be addressed by the authors, mostly related to the deep learning portion of the methodology.

Comments:

1.     Where is Equation (1) referenced on the text?

2.     On Section 3.2, what is the k representing in Equation (2)? Kernel (mask)? Please clarify in the document.

3.     In this same Section (3.2), please improve mentioning how the vertex lattice data is being encoded to serve as input for the DNN. In this same regard, please improve Figure 3, since the relevant part (the input) is not properly described with the Figure, it is important to know the specifics of the mapping.

4.     Since the main architecture tested was Resnet, please provide some information on why the authors selected this architecture with residual operation instead of traditional CNNs.

5.     Please extend the description of the residual block.

6.     It seems that the information from Table 1 is not mentioned/discussed in the text. Additionally, why the trainable data for each method is different? How was the train/test/val split done for each dataset?

7.     In Figure 4, authors are showing training accuracy, which is incorrect to measure the performance of the model, since the model can be overfitting (as could be from the graph after epoch 300). Please provide validation metrics instead.

8.     In Section 4, the authors state that increasing the network depth could led to longer training times, but the tested architectures are very shallow, how much time did it take on average to train? What GPUs did the authors used for training?

9.     Please provide training hyperparameters for the model besides epochs.

Author Response

(The authors gave the same response as above.)

Round 2

Reviewer 1 Report

As we carefully noticed, the authors were unable to respond to all of my comments. The literature review is inadequate. Again, we are searching for the quantum computing aspect in this research, but we have found no relationship between quantum computing and machine learning. Again, we provide you the opportunity to defend all of our remarks, and you will need to cite additional references that enhance your work with quantum computing.

https://doi.org/10.3390/e25020287

https://doi.org/10.1016/j.asej.2017.02.005

https://doi.org/10.3390/e25020287

https://doi.org/10.1007/s10825-017-0960-4

Restructuring is necessary. 

Author Response

Thank you again for your guidance and suggestions. In the newly submitted manuscript, we have revised and strengthened the references to quantum computing and machine learning, highlighted them with green text, and further improved the quality of the English writing.

Reviewer 2 Report

Authors replied to all the reviewer doubts. Manuscript can be approved in present form.

Please do a revision for minor typos in the document.

Author Response

Thank you again for your guidance. we have further improved our English writing and corrected minor typos in the article.